# Beyond Fine Tuning: A Modular Approach to Learning on Small Data

**Aryk Anderson**[‡]
Eastern Washington University
Cheney, Washington
`aryk.anderson@eagles.ewu.edu`

**Kyle Shaffer**,[*] **Artem Yankov**,[*] **Courtney D. Corley**[†] **& Nathan O. Hodas**[†]
Pacific Northwest National Laboratory
Washington, USA
`{kyle.shaffer,artem.yankov,court,nathan.hodas}@pnnl.gov`

## Abstract

In this paper we present a technique to train neural network models on small amounts of data. Current methods for training neural networks on small amounts of rich data typically rely on strategies such as fine-tuning a pre-trained neural network or the use of domain-specific hand-engineered features. Here we take the approach of treating network layers, or entire networks, as modules and combine pre-trained modules with untrained modules, to learn the shift in distributions between data sets. The central impact of using a modular approach comes from adding new representations to a network, as opposed to replacing representations via fine-tuning. Using this technique, we are able surpass results using standard fine-tuning transfer learning approaches, and we are also able to significantly increase performance over such approaches when using smaller amounts of data.

## 1 Introduction

Training generalizable models using only a small amount of data has proved a significant challenge in the field of machine learning since its inception. This is especially true when using artificial neural networks, with millions or billions of parameters. Conventional wisdom gleaned from the surge in popularity of neural network models indicates that extremely large quantities of data are required for these models to be effectively trained. Indeed the work from Krizhevsky et al. (2012) has commonly been cited as only being possible through the development of ImageNet (Russakovsky et al. (2015)). As neural networks become explored by practitioners in more specialized domains, the volume of available labeled data also narrows. Although training methods have improved, it is still difficult to train deep learning models on small quantities of data, such as only tens or hundreds of examples.

The current paradigm for solving this problem has come through the use of pre-trained neural networks. Bengio et al. (2012) were able to show that transfer of knowledge in networks could be achieved by first training a neural network on a domain for which there is a large amount of data and then retraining that network on a related but different domain via fine-tuning its weights. Though this approach demonstrated promising results on small data, these models do not retain the ability to function as previously trained. That is, these models end up fine tuning their weights to the new learning task, forgetting many of the important features learned from the previous domain.

The utility of pre-training models extends beyond training on small data. It is also used as an effective initialization technique for many complicated models (Jaderberg et al. (2015); Lakkaraju et al. (2014)). This, in addition to the continuing trend of treating specific network layer architectures as modular components to compose more advanced models (He et al. (2015); Larsson et al. (2016); Szegedy et al. (2015); Abadi et al. (2016)) informs our work as we seek to use pre-trained models as

---

[*]Seattle, WA
[†]Richland, WA; AA and NOH contributed equally

an architectural framework to build upon. Instead of overwriting these models and fine-tuning the internal representations to a specific task, we propose composing pre-trained models as modules in a higher order architecture where multiple, potentially distinct representations contribute to the task. With this approach, useful representations already learned are not forgotten and new representations specific to the task are learned in other modules in the architecture.

In this paper we present our neuro-modular approach to fine-tuning. We demonstrate how modules learn subtle features that pre-trained networks may have missed. We quantitatively compare traditional fine-tuning with our modular approach, showing that our approach is more accurate on small amounts of data ($<100$ examples per class). We also demonstrate how to improve classification in a number of experiments, including CIFAR-100, text classification, and fine-grained image classification, all with limited data.

## 2 RELATED WORK

Transferring knowledge from a source domain to a target domain is an important challenge in machine learning research. Many shallow methods have been published, those that learn feature invariant representations or by approximating value without using an instance's label (Pan & Yang (2010); Sugiyama et al. (2008); Pan et al. (2011); Zhang et al. (2013); Wang & Schneider (2014); Gong et al. (2016)). More recent deep transfer learning methods enable identification of variational factors in the data and align them to disparate domain distributions (Tzeng et al. (2014); Long et al. (2015); Ganin & Lempitsky (2014); Tzeng et al. (2015)). Mesnil et al. (2012) presents the Unsupervised and Transfer Learning Challenge and discusses the important advances that are needed for representation learning, and the importance of deep learning in transfer learning.Oquab et al. (2014) applied these techniques to mid-level image representations using CNNs. Specifically, they showed that image representations learned in visual recognition tasks (ImageNet) can be transferred to other visual recognition tasks (Pascal VOC) efficiently. Further study regarding the transferability of features by Yosinski et al. (2014) showed surprising results that features from distant tasks perform better than random features and that difficulties arise when optimizing splitting networks between co-adapted neurons. We build on these results by leveraging existing representations to transfer to target domains without overwriting the pre-trained models through standard fine-tuning approaches.

Long et al. (2015) developed the Deep Adaptation Network (DAN) architecture for convolutional neural networks that embed hidden representations of all task-specific layers in a reproducing kernel Hilbert space. This allows the mean of different domain distributions to be matched. Another feature of their work is that it can linearly scale and provide statistical guarantees on transferable features. The Net2Net approach (Chen et al. (2015)) accelerates training of larger neural networks by allowing them to grow gradually using function preserving transformations to transfer information between neural networks. However, it does not guarantee that existing representational power will be preserved on a different task. Gong et al. (2016) consider domain adaptation where transfer from source to domain is modeled as a causal system. Under these assumptions, conditional transferable components are extracted which are invariant after location-scale transformations. Long et al. (2016) proposed a new method that overcomes the need for conditional components by comparing joint distributions across domains. Unlike our work, all of these require explicit assumptions or modifications to the pre-trained networks to facilitate adaptation.

We note that while writing this paper, the progressive network architecture of Rusu et al. (2016) was released, sharing a number of qualities with our work. Both the results we present here and the progressive networks allow neural networks to extend their knowledge without forgetting previous information. In addition, Montone et al. (2015) discusses a semi-modular approach. Montone et al. also froze the weights of the original network, although it did not focus on the small data regime, where only a few tens of examples could be available. However, our modular approach detailed here focuses on leveraging small data to adapt to different domains. Our architecture also complements existing network building strategies, such as downloading pre-trained neural networks to then be fine-tuned for domain adaptation.

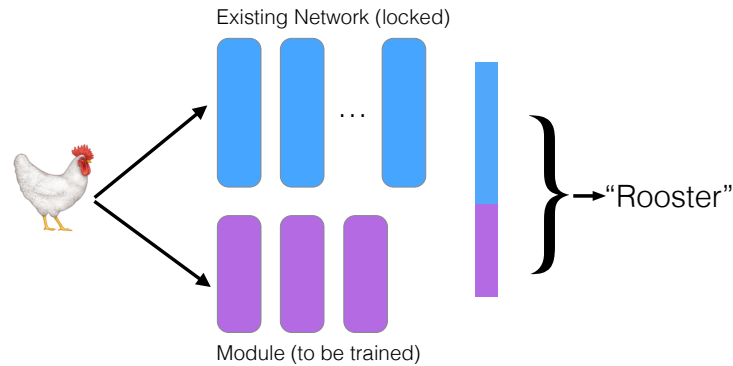

Figure 1: The modular approach to neural networks involves feeding data through one or more pre-existing neural networks as well as a new network, the module. The existing networks have their weights locked, so they will not be altered by the training process. Only the module weights are trained. The end result is a representation that adds a new representation to an existing representation without losing any information from the original network.

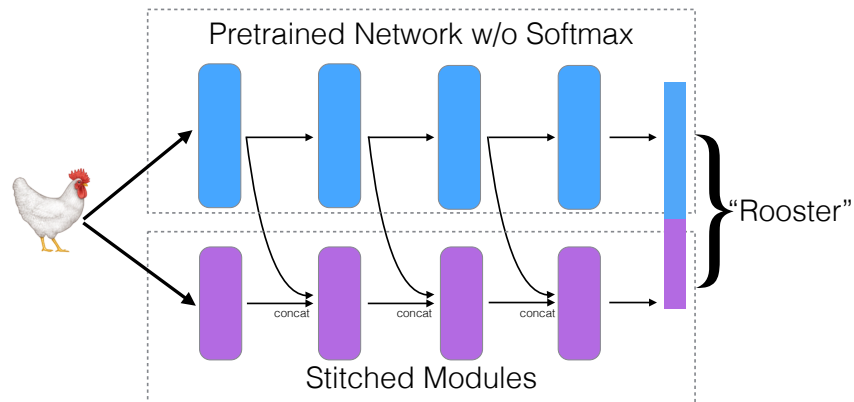

Figure 2: Modular networks do not simply need to be two models in parallel. Here, we present the stitched module approach. We insert a small neural network between each layer of the original network. This way, the modules explicitly receive information about the representations at each layer of the pre-trained network.

## 3 MODULAR ARCHITECTURE

Generically, modular neural networks are directed graphs of pre-trained networks linked together with auxiliary, untrained networks. Depicted in Fig. 1, one only trains the new components of the network. The architecture could take the form of simply placing two networks in parallel (the two-towers approach), shown in Fig. 1. In addition, the architecture could interleave the modules with the layers of the pre-trained network (the stitch approach), shown in Fig. 2

This allows the network as a whole to retain the original representational power of the pre-trained network. Thus, our modular approach bounds from below the performance of transfer learning. Here, we explore some of the properties of these modular architectures, including how they learn new representations and how they perform on small amounts of data.

### 3.1 LEARNED FILTERS

In the case of convolutional networks, we posit that adding modules to networks helps them learn new domains because the original modules contribute well-trained filters, allowing the untrained modules to learn more subtle features that may perhaps be more discriminating. Even slight regularization on the module network will encourage the network to avoid redundancy with the base network.

To visualize images that maximally stimulate each filter, we followed the approach of Zeiler & Fergus (2014). We set the objective function to be the activation of the filter we were querying. We then conducted back-propagation. Instead of using the gradients to alter the weights, we used the gradients at the input layer to alter the pixels themselves. We initialized with an image of noise smoothed with a Gaussian filter of radius 1. The gradient was normalized, so the input image, $X$ was updated according to

$$X_{t+1} = X_t + 0.01 * \nabla/|\nabla|,$$

where $\nabla$ is the induced gradient at the input layer. This was repeated 500 times, at which point the image largely had converged.

After training a simple neural network on MNIST with 3 convolutional layers, $(8 \times 8 \times 8) - maxpool2 - (8 \times 4 \times 4) - (8 \times 3 \times 3) - Dense128$, which was done using ADAM Kingma & Ba (2014) and augmenting the images with 10% shifts and zooms, we reached an accuracy of 98.8%. We then added an even simpler module to the neural network, $(4 \times 8 \times 8) - maxpool2 - (4 \times 4 \times 4) - (4 \times 3 \times 3) - Dense32$. This module is trained on the same input as the original model but it is tied together with the output features of the original model, as illustrated in Fig. 1. After training the module, the combined network achieves 99.2% accuracy. The models were intentionally kept small, with the original model only having 8 filters per layer, and the module only having 4 filters per layer.

As we can see in Fig. 3, the module does not learn filters that merely duplicate the original network. As is common, the first layer learns typical edge and orientation detectors, but the module is more sensitive to high-frequency diagonal components and details around the edge of the image. In the second layer, we see that the module is sensitive to diagonal components near the boundary. And the third layer shows that the module has indeed concentrated its effort on detecting strokes near the edge of the image. As we can see from inspecting Figure 3c, while the original network concentrated its efforts on the center of the images (as it should), the module was then able to focus more around the edges of the image and catch some of the mistakes made by the original network.

### 3.2 SMALL DATA

Although the modular approach can be used to extend and improve a network on its original task, its value comes from its ability to facilitate transfer learning. If a network has been trained on thousands or even millions of examples and hand-tuned for weeks or months, one would not want to throw away this valuable representational power by training the network with 100 examples from an out-of-domain dataset. Instead, the modular approach keeps the original, unaltered network, in addition to learning supplementary representations specific to the distribution of the new data.

This allows the modular approach to more robustly handle small data sets than naive fine-tuning. To demonstrate this, we trained a network on CIFAR-10 and used it to apply to CIFAR-100 for varying

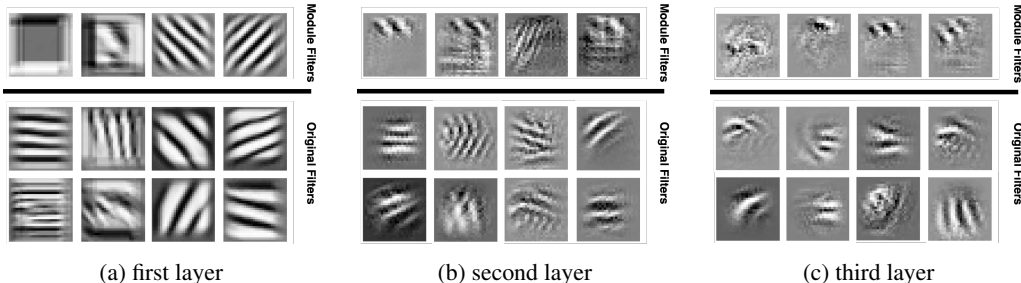

|  (a) first layer  |  (b) second layer  |  (c) third layer  |

Figure 3: After training a vanilla CNN on MNIST, images that maximally stimulate each filter are shown on the bottom rows. Images that maximally stimulate the auxiliary module network, trained on the same data to supplement the original network, are shown on the top.

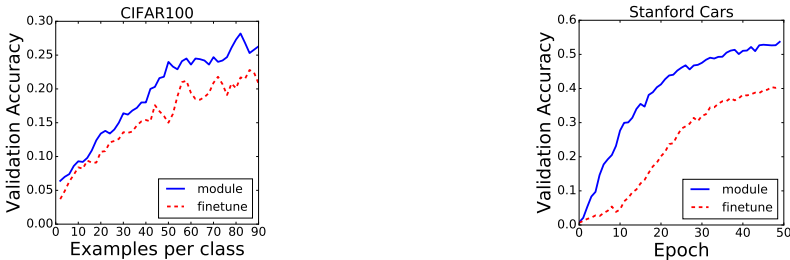

(a) Comparison of modular approach vs. fine-tuning based on amount of training data

(b) Comparison of validation accuracy on the Stanford Cars data. Training on full data

Figure 4: By explicitly preserving the original representation learned on pre-trained net, the module is able to learn more robust features using fewer examples than naive fine-tuning.

amounts of training data. The CIFAR-10 network was trained until it was 88.9% accurate, using the network in He et al. (2016) with 3 residual units, for a total of 28 layers.

We then compared two approaches. For the first approach, we simply fine tuned the CIFAR-10 network by using training data from the CIFAR-100 dataset and replacing the final softmax. Second, we froze the original CIFAR-10 network and added an identical copy as a module, which would be trained on the same batches of data as the first approach. That is, we have: Network 1 – fine-tuning the base network and Network 2 – freezing the base network and fine-tuning a module. This doubles the amount of weights in the second network, but Network 1 and Network 2 have an identical number of weights to be trained and those weights have the same starting value. More formally, we present these two approaches in equations 1 and 2 below.

$$y_{ft} = \text{softmax}(NN(x; w_0 = \{C_{10}\})) \tag{1}$$

$$y_{mod} = \text{softmax}([NN(x; w^\star = \{C_{10}\}), NN(x; w_0 = \{C_{10}\})]) \tag{2}$$

where $y_{ft}$ denotes predictions made from a fine-tuned network and $y_{mod}$ denotes predictions made from our modular architecture. $NN$ denotes the neural network without softmax activation trained on CIFAR-10, and $w_0$ is the initialization of the weights, which are learned from training on CIFAR-10, i.e. $w_0 = \{C_{10}\}$. Note that in our modular architecture pre-trained weights are locked as denoted by $w^\star = \{C_{10}\}$ in Equation2, i.e., $\nabla_w NN(w^\star) \equiv 0$.

To train, we used the ADAM optimization algorithm (Kingma & Ba (2014)). We added an activity L2 regularization of $1e^{-6}$ to the module to help break degeneracy. We used batches of 200, where each batch contained two images per class. Each batch was iterated over five times, before the next batch was used. This iteration allowed simulating multiple epochs over small data. We recorded the results of the performance on the test set after each batch, in Fig. 4a.

We observe that for all amounts of training data, but particularly for small amounts of training data, the modular approach outperforms traditional fine-tuning. Of course, we chose to make the module

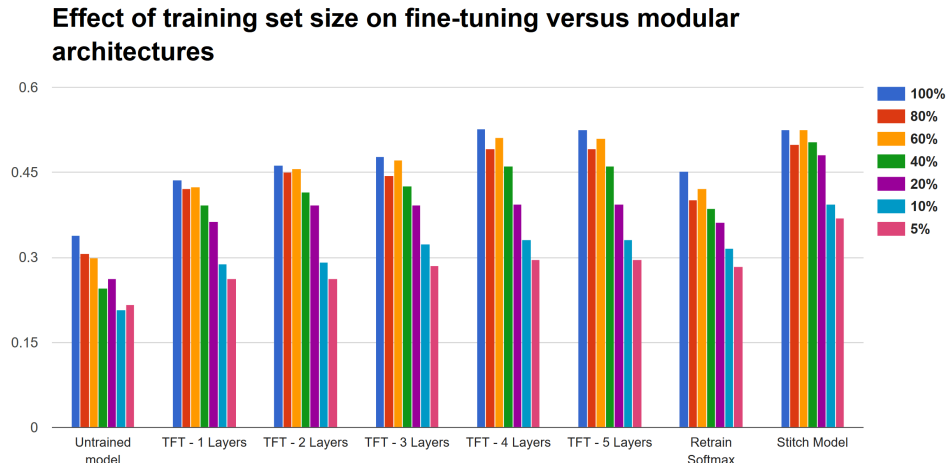

Figure 5: Comparison of fine tuning vs the stitched module approach. TFT stands for 'Traditional Fine Tuning.' and the number of layers fine-tuned is indicated. Notice that our modular approach outperforms fine-tuning for all amounts of training data. The modular approach's benefit over fine-tuning increases as the amount of available training data decreases.

a complete copy of the original CIFAR-10 network. This ensured we could compare with the same number of weights, same initialization, same data, etc. Further research will certainly reveal more compact module networks that outperform our example.

## 4 EXPERIMENTS

### 4.1 TRANSFER LEARNING FROM CIFAR-10 TO CIFAR-100 WITH STITCH NETWORKS

To investigate the effectiveness of modular networks for transfer learning, we explore a second example of transfer learning from CIFAR-10 in order to model CIFAR-100. As we were able to show above, a modular network is able to outperform traditional fine-tuning because it learns additional features that may complement those captured by the pre-trained model. However, there is no reason why a module needs to only accept input from the input layer nor a reason why it needs to send its output directly to the softmax layer. Here, we describe stitch networks, where the modules are actually interwoven with the original network.

We believe that in modular networks, the untrained module learns representations that capture the difference from the original distribution of data to the distribution of data under the new task. Expanding upon this idea, instead of learning the shift in distributions only at the softmax layer as with our other modular networks, we integrate the signal from the learned modules much more tightly with the paired untrained modules by using a Stitch Network. Use of the Stitch Network allows for the model to learn to correct the distribution difference after each transformation made by the learned module, shown in Fig. 2.

The stitch network we explore is comprised of layer pairs between a single learned and unlearned module. The learned module is a five layer convolutional neural network where the first two layers are 3x3 convolutional layers with 32 filters, followed by max pooling and two more 3x3 convolutions with 64 filters. The convolutions are followed by a fully connected layer with 512 outputs and finally a softmax for classification. This model is pre-trained on CIFAR-10 then stripped of the softmax layer, has its weights locked and is then used as the learned module for the stitch network. The untrained module is composed in a similar fashion, with four 3x3 convolutions with maxpooling, and a fully connected layer each with 1/4 the number of outputs as the corresponding pre-trained layers. The outputs of each layer pair are concatenated and fed as the input for each proceeding layer of the untrained module. Both modules feed into the final softmax layer of the composite network which then classifies over the new data set. A sketch of this is shown in Fig. 2

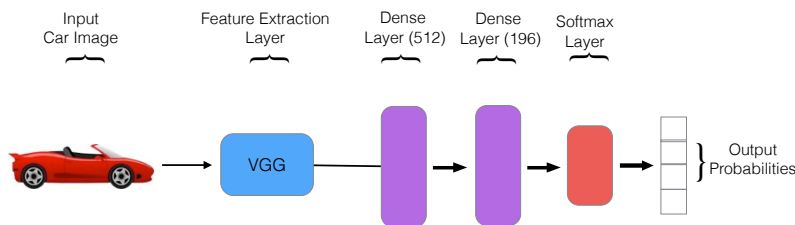

Figure 6: Network architecture used for Stanford Cars fine-tuned model.

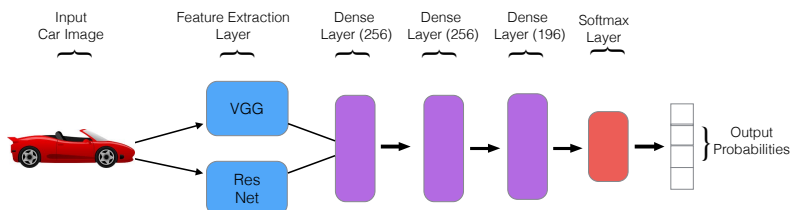

Figure 7: Network architecture used for Stanford Cars module model. Note, the ResNet used is identical to the one describe in He et al. (2015).

We train our entire composite model on a randomly selected subset of ten CIFAR-100 classes. We compare the accuracy over the validation set of the selected classes against traditional fine-tuning using only the learned module, as well as against an uninitialized version of the learned module. We were additionally interested in comparing across all models the effect of limiting the amount of data available for training. We repeat the experiment with the same subset of classes, varying the amount of available training data such that the networks are shown only a fraction of each class for training. Note that there are 500 available training examples per class in CIFAR-100.

We find by using the stitch network, we are able to match or improve upon classification results (Fig. 5) obtained using traditional fine-tuning over a pre-trained model. We also outperform training from scratch, regardless of the amount of training data used. We note that we find significant gains over traditional methods as the number of available training examples drops below 200 examples per class.

## 4.2    STANFORD CARS DATA SET

The Stanford Cars data set (Krause et al. (2013)), which features 16,185 images of 196 classes of cars, is an example of a data set for fine-grained categorization. Rather than train a classifier to distinguish between fundamentally different objects like horses and planes, as required in the Large Scale Visual Recognition Challenge (Russakovsky et al. (2015)), fine-grained categorization requires the classifier to learn subtle differences in variations of the same entity. For example, a classifier trained on the Stanford Cars data set would have to learn distinguishing features between a BMW X6 SUV from 2012 and an Isuzu Ascender SUV from 2008.

In this research two models are trained on the Stanford Cars data set. Both models utilize a transfer learning approach by leveraging the non-fully connected output from the VGG16 model (Simonyan & Zisserman (2014)). The "fine-tuned" model passes the VGG16 features to a fully connected layer of length 512 followed by a softmax layer of length 196, as seen in Fig. 6. Gradient descent via RMSPROP is used to train the dense layers. The "module" model merges the fixed VGG16 features with a ResNet (He et al. (2015)) model, whose output is then fed to two consecutive dense layers of length 256 capped by a softmax layer of length 196. The module model architecture is shown in Fig. 7. Again, RMSPROP is used to train ResNet and post-merge dense layer weights, but the VGG features are unchanged.

As seen in Fig. 4b, after 50 epochs the module model appears to significantly outperform the fine-tuned model in validation accuracy. However, it should be noted that while the module model carries

19,537,990 trainable parameters the fine-tuned model only has 12,946,116 parameters. Furthermore, no hyperparameter optimization is performed on either model.

## 4.3 Module for LSTM text classification

We further investigate the effects of our modular network approach by applying this method to a different modeling problem - text classification. Similar to image data, text represents an unstructured data-type that often exhibits long-term and interconnected dependencies within the data that are difficult to model with simpler classifiers. Whereas in the case of images neighboring pixels may represent semantically related concepts or objects, in text words may exhibit long-term semantic or syntactic dependencies that can be modeled sequentially. These characteristics make text classification particularly well-suited to recurrent neural networks such as long short-term memory (LSTM) networks, but these learning methods typically require a great deal of data to be learned efficiently and to avoid overfitting.

To test our methodology, we evaluate a modular recurrent network against two individual recurrent neural networks on the IMDB sentiment dataset. Previous work has shown deep learning methods to be effective at sentiment classification performance on this dataset (Maas et al. (2011)), however we add to this past work by presenting an analysis that demonstrates the effectiveness of modular networks in the case of extremely small training sets. To this end, we sample only 500 training examples from the original 25,000 available in the full training set, and evaluate on the full 25,000 validation examples. We use the same 500 training examples for each model evaluated in our experiments for consistency, and report accuracy for each model on the full validation set.

We evaluate three models in our text-classification experiments, two of which are individual recurrent networks and the final which is our modular recurrent network. The first model consists of three layers - an initial layer that projects sequences of words into an embedding space, a second LSTM layer with 32 units, and a final sigmoid layer for computing the probability of the text belonging to the positive class. Our second model is identical to the first except that we fix the weights of the embedding layer using pre-trained GloVe word vectors[1]. In particular, we use 100-dimensional vectors computed from a 2014 version of Wikipedia.

Finally, we detail our modular network, which leverages both individual recurrent neural networks described above. To construct our modular network, we take the embedding and LSTM layers from our individual networks, and concatenate the output of both LSTM layers into a single tensor layer in the middle of our modular network. Additionally, we modify the output of each of these component LSTM layers by forcing each to output a weight matrix that tracks the state of the LSTM layer across all timesteps throughout the dataset. In this way, we seek to fully leverage the sequential dependencies learned by this layer, and this method outperforms the simpler alternative method of simply outputting the final state of each of the LSTM layers. We then feed this concatenated layer to a gated recurrent unit (GRU) layer with a sigmoid activation function for calculation of class probabilities. We experimented with an LSTM and densely connected layers after the tensor concatenation layer, but found best performance with the GRU. All models were optimized with the ADAM algorithm, and trained for 15 epochs. An outline of this architecture can be seen in Figure 8.

Here, we report results for our classification experiments with the three networks described above. We see an accuracy of 61.9% for our first model which is trained directly from the data without any pre-training. This is significantly lower than previously reported results, however we are training on only 2% of the available data to test our method's application to small training sets. We see slightly better performance in terms of accuracy (64.9%) from our second model initialized with GloVe vectors. This seems to indicate that despite being trained on more formally written language in Wikipedia, these vectors can still boost performance on a task modeling text that is inherently subjective and opinion-based. Finally, we see an accuracy of 69.6% from our modular network, an increase of almost 5% accuracy over the next best performing model. Because weight initializations of recurrent networks can greatly affect model performance, we ran the classification experiments with our modular network 10 times, and report the average accuracy across these 10 runs. As can be seen here, our modular approach improves on the best performing individual network suggesting that

---

[1]http://nlp.stanford.edu/projects/glove/

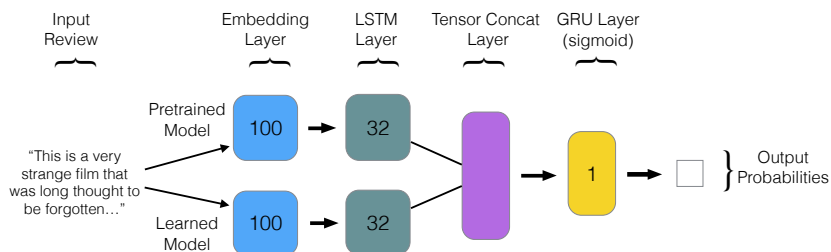

Figure 8: Diagram of architecture for our modular recurrent text classification network. Dimensionality for embedding layers and number of units for all other layers are given in boxes denoting those layers.

| MODEL | BM | PM | MM |
|---|---|---|---|
| ACCURACY (%) | 61.9 | 64.9 | **69.6** |

Table 1: Accuracy results for text classification experiments, using only 500 training examples. Results are shown for the baseline model (**BM**), pre-trained (GloVe) model (**PM**) and modular model (**MM**).

this approach is useful in the domain text classification, and that our modular approach overcomes the poor performance shown by one of its component models.

## 5 CONCLUSIONS

We have presented a neuro-modular approach to transfer learning. By mixing pre-trained neural networks (that have fixed weights) with networks to be trained on the specific domain data, we are able to learn the shift in distributions between data sets. As we have shown, often the new modules learn features that complement the features previously learned in the pre-trained network. We have shown that our approach out-performs traditional fine-tuning, particularly when the amount of training data is small – only tens of examples per class. Further research will explore more efficient architectures and training strategies, but we have demonstrated that our approach works well for MNIST, CIFARs, the Stanford Cars dataset, and IMDB sentiment. Thus, the modular approach will be a valuable strategy when one has a large pre-trained network available but only a small amount of training data in the transfer task.

ACKNOWLEDGMENTS

This work was supported by the US Government.

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
