# Peer review of "Beyond Fine Tuning: A Modular Approach to Learning on Small Data"

_ICLR 2017 — rejected_

[Official Review · AnonReviewer1 · rating 6 · confidence 5 · 14 Dec 2016]
**No Title**

This paper presents a new technique for adapting a neural network to a new task for which there is not a lot of training data. The most widely used current technique is that of fine-tuning. The idea in this paper is to instead learn a network that learns features that are complementary to the fixed network. Additionally, the authors consider the setting where the new network/features are “stitched” to the old one at various levels in the hieararchy, rather that it just being a parallel “tower”. 

This work is similar in spirit (if not in some details) to the Progressive Nets paper by Rusu et al, as already discussed. The motivations and experiments are certainly different so this submission has merit on its own.

The idea of learning a “residual” with the stitched connnections is very similar in spirit to the ResNet work. It would be nice to compare and contrast those approaches.

I’ve never seen a batch being used 5 times in a row during training, does this work better than just regular SGD?

In Figure 5 it’d be nice to label the y-axis. That Figure would also benefit from not being a bar chart, but simply emulating Figure 4, which is much more readable!

Figure 5 again: what is an untrained model? It’s not immediately obvious why this is a good idea at all. Is TFT-1 simply fine-tuning one more layer than “Retrain Softmax”?

I think that the results at the end of section 3 are a bit weak because of usage of a big network. I would definitely like to see how the results change if using a smaller net.

The authors claim throughout the paper that the purpose of the added connections and layers is to learn *complementary* features and they show this with some figures. The latter are a convinving evidence, but not proof or guarantee that this is what is actually happening. I suggest the authors consider adding an explicit constraint in their loss that encourages that, e.g. by having a soft orthogonality constraing (assuming one can project intermediate features to some common feature dimensionality). The usage of very small L2 regularization maybe achieves the same thing, but there’s no evidence for that in the paper (in that we don’t have any visualizations of what happens if there’s no L2 reg.).

One of the big questions for me while reading the paper was how would an ensemble of 2 pre-trained nets would do on the tasks that the authors consider. This is especially relevant in the cars classification example, where I suspect that a strong baseline is that of fine-tuning VGG on this task, fine-tuning resnet on this task, and possibly training a linear combination of the two outputs or just averaging them naively.

Disappointing that there are no results in figure 4, 5 and 8 except the ones from this paper. It’s really hard to situate this paper if we don’t actually know how it compares to previously published results.


In general, this was an interesting and potentially useful piece of work. The problem of efficiently reusing the previously trained classifier for retraining on a small set is certainly interesting to the community. While I think that this paper takes a good step in the right direction, it falls a bit short in some dimensions (comparisons with more serious baselines, more understanding etc).

[Official Review · AnonReviewer3 · rating 6 · confidence 2 · 16 Dec 2016]
**No Title**

This paper proposed to perform finetuning in an augmentation fashion by freezing the original network and adding a new model aside it. The idea itself is interesting and complements existing training and finetuning approaches, although I think there are a few baseline approaches that can be compared against, such as:

(1) Ensemble: in principle, the idea is similar to an ensembling approach where multiple networks are ensembled together to get a final prediction. The approach in Figure 1 should be compared with such ensemble baselines - taking multiple source domain predictors, possibly with the same modular setting as the proposed method, and compare the performance.

(2) comparison with late fusion: if we combine the pretrained network and a network finetuned from the pretrained one, and do a late fusion?

Basically, I think it is a valuable argument in section 3.2 (and Figure 4) that finetuning with a small amount of data may hurt the performance in general. This builds the ground for freezing a pretrained network and only augmenting it, not changing it. I agree with the authors on this argument, although currently other than Figure 4 there seem to be little empirical study that justifies it.

It is worth noting that Figure 3 seems to suggest that some of the module filters are either not converging or are learning unuseful features - like the first two filters in 3(a).

Overall I think it is an interesting idea and I would love to see it better developed, thus I am giving a weak accept recommendation, but with a low confidence as the experiments section is not very convincing.

[Official Review · AnonReviewer2 · rating 4 · confidence 4 · 17 Dec 2016 (modified: 24 Jan 2017)]
**needs stronger experimental validation**

This paper proposes a method of augmenting pre-trained networks for one task with an additional inference path specific to an additional task, as a replacement for the standard “fine-tuning” approach.

Pros:
-The method is simple and clearly explained.
-Standard fine-tuning is used widely, so improvements to and analysis of it should be of general interest.
-Experiments are performed in multiple domains -- vision and NLP.

Cons:
-The additional modules incur a rather large cost, resulting in 2x the parameters and roughly 3x the computation of the original network (for the “stiched” network).  These costs are not addressed in the paper text, and make the method significantly less practical for real-world use where performance is very often important.

-Given these large additional costs, the core of the idea is not sufficiently validated, to me.  In order to verify that the improved performance is actually coming from some unique aspects of the proposed technique, rather than simply the fact that a higher-capacity network is being used, some additional baselines are needed:
(1) Allowing the original network weights to be learned for the target task, as well as the additional module.  Outperforming this baseline on the validation set would verify that freezing the original weights provides an interesting form of regularization for the network.
(2) Training the full module/stitched network from scratch on the *source* task, then fine-tuning it for the target task.  Outperforming this baseline would verify that having a set of weights which never “sees” the source dataset is useful.

-The method is not evaluated on ImageNet, which is far and away the most common domain in which pre-trained networks are used and fine-tuned for other tasks.  I’ve never seen networks pre-trained on CIFAR deployed anywhere, and it’s hard to know whether the method will be practically useful for computer vision applications based on CIFAR results -- often improved performance on CIFAR does not translate to ImageNet.  (In other contexts, such as more theoretical contributions, having results only on small datasets is acceptable to me, but network fine-tuning is far enough on the “practical” end of the spectrum that claiming an improvement to it should necessitate an ImageNet evaluation.)

Overall I think the proposed idea is interesting and potentially promising, but in its current form is not sufficiently evaluated to convince me that the performance boosts don’t simply come from the use of a larger network, and the lack of ImageNet evaluation calls into question its real-world application.

===============

Edit (1/23/17): I had indeed missed the fact that the Stanford Cars does do transfer learning from ImageNet -- thanks for the correction.  However, the experiment in this case is only showing late fusion ensembling, which is a conventional approach compared with the "stitched network" idea which is the real novelty of the paper.  Furthermore the results in this case are particularly weak, showing only that an ensemble of ResNet+VGG outperforms VGG alone, which is completely expected given that ResNet alone is a stronger base network than VGG ("ResNet+VGG > ResNet" would be a stronger result, but still not surprising). Demonstrating the stitched network idea on ImageNet, comparing with the corresponding VGG-only or ResNet-only finetuning, could be enough to push this paper over the bar for me, but the current version of the experiments here don't sufficiently validate the stitched network idea, in my opinion.

[Final Decision · Program Chairs · 06 Feb 2017]
**ICLR committee final decision**

The method was developed to provide an alternative for fine-tuning by augmenting a pre-trained network with new capacity. The differential from other related methods is low, and the evaluated baselines were not well-chosen, so this is not a strong submission.